# Random Forests Highlight the Combined Effect of Environmental Heavy Metals Exposure and Genetic Damages for Cardiovascular Diseases

**Alfonso Monaco** [1] , **Antonio Lacalamita** [2] , **Nicola Amoroso** [1,3,*] , **Armando D'Orta** [4] , **Andrea Del Buono** [4] , **Francesco di Tuoro** [4] , **Sabina Tangaro** [1,5] , **Aldo Innocente Galeandro** [6,†] **and Roberto Bellotti** [1,7,†]

1   Istituto Nazionale di Fisica Nucleare (INFN), Sezione di Bari, 70125 Bari, Italy; alfonso.monaco@ba.infn.it (A.M.); sonia.tangaro@ba.infn.it (S.T.); roberto.bellotti@ba.infn.it (R.B.)
2   National Institute of Gastroenterology S. de Bellis, Research Hospital, Castellana Grotte, 70013 Bari, Italy; antonio.lacalamita@irccsdebellis.it
3   Dipartimento di Farmacia—Scienze del Farmaco, Università degli Studi di Bari Aldo Moro, Via A. Orabona 4, 70125 Bari, Italy
4   Fondazione Ddclinic Institute Research, Via Fratelli Bandiera 35, 81100 Caserta, Italy; armandodorta@gmail.com (A.D.); delbuonosm@gmail.com (A.D.B.); dituoro.f@gmail.com (F.d.T.)
5   Dipartimento di Scienze del Suolo, Della Pianta e degli Alimenti, Università degli Studi di Bari Aldo Moro, Bari, Via G. Amendola 165, 70125 Bari, Italy
6   Unità di Ricerca di Tecnologia Medica PST, Università degli Studi di Bari Aldo Moro, Str. P.le per Casamassima km.3, Valenzano, 70010 Bari, Italy; aldogaleandro@libero.it
7   Dipartimento Interateneo di Fisica M. Merlin, Università degli Studi di Bari Aldo Moro, Via G. Amendola 173, 70125 Bari, Italy
*   Correspondence: nicola.amoroso@uniba.it
†   Equal last author contribution.

**Abstract:** Heavy metals are a dangerous source of pollution due to their toxicity, permanence in the environment and chemical nature. It is well known that long-term exposure to heavy metals is related to several chronic degenerative diseases (cardiovascular diseases, neoplasms, neurodegenerative syndromes, etc.). In this work, we propose a machine learning framework to evaluate the severity of cardiovascular diseases (CVD) from Human scalp hair analysis (HSHA) tests and genetic analysis and identify a small group of these clinical features mostly associated with the CVD risk. Using a private dataset provided by the DD Clinic foundation in Caserta, Italy, we cross-validated the classification performance of a Random Forests model with 90 subjects affected by CVD. The proposed model reached an AUC of $0.78 \pm 0.01$ on a three class classification problem. The robustness of the predictions was assessed by comparison with different cross-validation schemes and two state-of-the-art classifiers, such as Artificial Neural Network and General Linear Model. Thus, is the first work that studies, through a machine learning approach, the tight link between CVD severity, heavy metal concentrations and SNPs. Then, the selected features appear highly correlated with the CVD phenotype, and they could represent targets for future CVD therapies.

**Keywords:** machine learning approaches; human scalp hair analysis (HSHA); cardiovascular diseases (CVD)

## 1. Introduction

Heavy metals, such as lead, mercury, aluminium, arsenic, and others, pose severe threats to human health [1]. In fact, their absorption by human bodies, in several heterogeneous ways and doses, can perturb and eventually damage the normal health balance; besides, they can affect human health at different scales, ranging from cellular DNA to anatomical structures such as the nervous system [2]. Heavy metals have a strong penetrating power allowing them, in some cases, to overcome the different protective structures of the human body, such as the blood–brain barrier, restricting, amongst other functions, the

access of pathogens into the cerebrospinal fluid or the placenta which can protect the fetus from maternal toxins [3].

The effects of the so-called external poisons (or xenobiotics) are not new to medical science. In fact, heavy metals have been present in a variety of natural sources, including air, drinking water and food, and have been since at least the dawn of human life. Thanks to evolution, human beings have elaborated a series of techniques and processes of defense. However, since the Industrial Revolution, concentrations of xenobiotics in the environment have increased dramatically [4]. Several studies by the Agency for Toxic Substances and Disease Registry (ATSDR) stated that different chemicals containing heavy metals are released into the environment and interact with entire biological ecosystems [5–11].

Depending on the concentration and exposure time, heavy metal damage can be divided into three different increasing levels: (i) toxic overload, a slow accumulation of damaging substances; (ii) chronic poisoning, a regular exposure to toxic but not lethal concentrations of metals; and (iii) acute poisoning. The concentration, the exposure time and especially the type of heavy metal determine the toxicity degree and affect the biological processes which are involved. Xenobiotics can take over the biological tasks that in a normal metabolism are occupied by essential nutritional minerals, thus creating an obstacle to the normal development of vital processes. Moreover, they can significantly increase the production of free radicals that alter the structural and functional integrity of healthy cells making the organism more vulnerable to degenerative pathologies such as neoplasia, cardiovascular diseases, Alzheimer's and early aging caused by DNA damage [12]. Another factor that influences heavy metals' toxicity is their non-biodegradability; therefore, once released in the environment, they tend to bio-concentrate and make their way to humans. For example, mercury compounds, once released in the sea as industrial waste, can change into methylmercury due to bacteria metabolism, thus representing a very strong toxic substance for human neurons [13].

Heavy metals in the human body damage DNA causing the formation of reactive oxygen species and alterations in cells. They produce mutation and induce an inhibition in the DNA repair mechanism contributing to genetic damage and double strand breaks (DSBs) [14,15]. For example, elements like arsenic, nickel and cadmium are considered among the activators of mutagenic changes in cell [16]. Furthermore, oxidative deterioration of biological macromolecules is mainly caused by the binding of heavy metals to DNA and nuclear proteins [14]. Heavy metals can damage not only the genetic makeup of somatic cells but also of germ cells. Benoff et al. found that environmental cadmium exposures could reduce human male sperm concentration and motility [17].

Cardiovascular disease (CVD) is an increasing global health problem and represents the number one cause of death in the occidental world [18]. In 2017, the annual world mortality rate of CVD was approximately 18 million [19] and it is expected to reach 23.6 million deaths by 2030 [20]. CVD can be caused by a coincidence of different risk factors that comprise genetic, behavioural and environmental factors in the broadest sense. Recent studies have shown that exposure to heavy metals increases risks of diabetes and hypertension [21,22]. Some essential (Co, Cu, Cr, Ni, Se) and toxic metals (As, Cd, Pb, and Hg) for the human body may also increase the risk of CVD through endocrine disruption [23]. Few epidemiologic studies in the US have highlighted a low and moderate association between arsenic exposure contained in drinking water and cardiovascular risks [24,25]. Furthermore, Wang et al. [26] reported that carotid atherosclerosis is linked with ingested inorganic arsenic. The association between blood cadmium and blood pressure is mentioned by Gallagher and Meliker [27] while Salonen et al. [28] found that an accumulation of mercury in the body is linked with an excess risk of acute myocardial infarction, just to name a few works. In general, in the last two decades evidence about the role of exposure to environmental heavy metals in CVD risk has rapidly increased [29]; however, the true mechanism by which heavy metals affect cardiovascular risk factors still remains quite unknown and difficult to identify [30]. Due to the complex physiology of CVD, patients differ for some pathological aspects of disease such as: the time of

onset, dynamics, and outcomes. This disease-related complexity makes the relevance of environmental factors difficult to quantify and study despite a growing understanding of the genetic, protein and molecular mechanisms involved in CVD [30].

Accordingly, in this work we investigate the association between single nucleotide polymorphisms (SNPs) and pathological conditions related to environmental heavy metals focusing on cardiovascular diseases.

Human scalp hair analysis (HSHA) is an alternative to blood and urine testing used to monitor the environmental and occupational pollution [31–33]. It is highly reliable and accurate even when heavy metal concentrations are very low. It is accepted by the US Food and Drugs Administration and by several International Agencies and organizations such as the Environmental Protection Agency (EPA) and World Health Organization (WHO). In this context, the development of accurate diagnostic decision support systems, based on HSHA tests, for the early detection of cardiovascular diseases to monitor patient conditions and assess the severity of symptoms is of paramount importance. As far as we know, we are the first to use a machine learning approach to correlate HSHA data with polymorphisms that specifically affect the coagulation cascade and the integrity of the vascular endothelium. In this context, machine learning methods can also help to clarify the connection between environmental factors, in particular heavy metal exposure and CVD risks, since these computational techniques do not require any a priori assumptions on the model and allow to highlight even very complex relationships. For this purpose, we enrolled a cohort of 90 patients affected by CVD at the DD Clinic foundation (https://www.ddclinicfoundation. eu/, accessed on 31 August 2021) of Caserta, Italy. The DD Clinic foundation finds its mission within the Integrated Personalized Medicine domain, specifically, aiming at the elaboration of cellular detoxification protocols from heavy metals with a particular attention to functional foods and nutraceutical supports to slow down or block the specific genetic predispositions of chronic degenerative diseases. We examined HSHA tests of patients involved and we grouped them into three clinical classes associated with three distinct levels of clinical risk. We used a machine learning approach, to evaluate which are the most important heavy metals associated to the aforementioned three clinical groups and their predictive power. In this way, the study of specific heavy metals particularly linked to CVD risk can facilitate the identification of true biological mechanisms through which environmental aspects and heavy metals influence cardiovascular risk factors and can shed light on the genetic basis of the disease. Thus, according to the best practices of the explainable artificial intelligence framework, we tried to assess transparency, justification, informativeness and uncertainty related to the proposed framework.

Based on the hypothesis that heavy metal concentrations and SNPs can accurately model the severity of CVD, the proposed framework provides a tool for remotely monitor and, eventually, diagnose the risk for cardiovascular risky conditions, thus representing an opportunity to widen the available possibilities in the telemedicine domain. Our findings could also pave the way for future therapies and the development of diagnostic tools to better characterize the diseases caused by the accumulation of heavy metals.

## 2. Materials and Methods

### 2.1. Data Description

We analyzed a dataset collected by the DD Clinic foundation of Caserta in Italy between September 2014 and February 2020. The DD Clinic foundation is a research institute and consists of 13 sections distributed throughout the Italian territory. The main research activities, carried out in collaboration with universities and public and/or private health organisations, are in the field of oncological, neurodegenerative and cardiovascular diseases. In this study, 90 patients with a CVD diagnosis were enrolled; they signed an informed and underwent a HSHA test to evaluate the concentrations of 27 heavy metals in their hair. The analysed subjects are Caucasians (55 males and 39 females) with median age of $53.5 \pm 6.4$ years.

In addition, every subject underwent a genomic analysis from which we selected SNPs of 25 genes linked to a CVD phenotype. Specifically, the genetic analysis is a nutrigenomic test and consists of a salivary swab to highlight genetic mutation that can cause metabolism alterations and the onset of several cardiovascular diseases. We implemented an allenic discrimination analysis using polymerase chain reaction (PCR) [34]. The enrolled subjects lived near the so called 'Land of Fire' (in Italian: Terra dei Fuochi)—a defined geographical area around the cities of Salerno and Naples in Campania. This area contains the largest illegal waste dump in Europe [35]. The 52 features used in our analysis (27 heavy metals and 25 SNPs) are listed in Table 1. According to the patients' clinical conditions, a three-level risk index was assigned to each subject: moderate (MR), high (HR) and very high risk (VHR). These levels are characterized by the same pathologies (heart attack, thrombosis, stroke, etc.), the difference is linked to the severity of the clinical conditions in terms of disability and risk of mortality. Further information is reported in Table 2. Specifically, the analysed dataset was composed of:

- 34 subjects in Class MR;
- 18 subjects in Class HR;
- 38 subjects in Class VHR.

**Table 1.** List of analysed heavy metals and selected SNPs used as features in our analysis. The heavy metal concentrations are extracted by means of TMA hair tests.

| Heavy Metals | | | |
|---|---|---|---|
| Aluminum | Arsenic | Barium | Beryllium |
| Bismuth | Boron | Cadmium | Calcium |
| Chromium | Cobalt | Copper | Iron |
| Lead | Lithium | Magnesium | Manganese |
| Mercury | Molybdenum | Nickel | Phosphorus |
| Platinum | Selenium | Sodium | Thallium |
| Titanium | Vanadium | Zinc | |
| **Polymorphisms** | | | |
| IL10 A1082G | TNF α/G-308A | CBS T1080C | IL-1βC-551T |
| IL4R 190A/G | BCO1 | VDR | ACE |
| IL-1α-899 (+4845) C>T | GSTM1 | IL-6 (-174)G/C | BHMT (GA AA AG) |
| MTFR CBS C6997 | MTR A2756G | MTR A66G | BHMT (GG AG AA) |
| IL6 634C/G and 174C/G | IL6R > Asp358Ala; rs2228145 A>C | MNSOD2 | MNSOD3 |
| MTHFR C677T | MTHFR A1298C | IFNG 874T/A | AHCY 257A/G |
| CAT C262T | | | |

**Table 2.** Clinical conditions linked to the three considered classes. In particular each level risk is characterized by different values of ESC SCORE RISK [36] and low-density lipoproteins (LDL) cholesterol.

| Class | Clinical Status |
|---|---|
| *MR* | ESC SCORE RISK < 5% and LDL threshold < 115 mg/dL |
| *HR* | Patients with familial dyslipidemia or severe hypertension, diabetics without cardiovascular risk factors and without organ damage and patients with moderate chronic renal failure (GFR > 30–59 mL/min/1.73 m$^2$). 5% < ESC SCORE RISK < 10% and LDL threshold < 100 mg/dL |
| *VHR* | Patients with documented cardiovascular disease (from coronary angiography, stress echocardiography, radionuclide imaging, ultrasound evidence of carotid plaque), previous myocardial infarction, previous acute coronary syndrome, previous coronary revascularization surgery with aorto-coraric bypass or percutaneous transluminal coronary angioplasty or peripheral, previous ischemic stroke and peripheral arterial disease, diabetic with one or more cardiovascular risk factors and/or markers of organ damage (and microalbuminuria) and with severe renal insufficiency (GFR < 30–59 mL/min/1.73 m$^2$). ESC SCORE RISK > 10% and LDL threshold < 70 mg/dL |

All patient data were anonymised. No data or elements attributable to patients were stored.

### 2.2. Methods

The goal of this study is the development of a diagnostic decision support system to evaluate the existence of a relationship between heavy metals and the inherent genetic mutations with the severity of associated CVD. We hypothesize here that the base of knowledge consisting of heavy metal concentrations and SNPs can accurately predict the severity of CVD. The proposed approach consists of four main steps: (i) data sampling; (ii) comparison of three learning models to choose the best performing one; (iii) feature selection; (iv) diagnosis through the most performing learning model. The data analysis procedure is summarized in Figure 1.

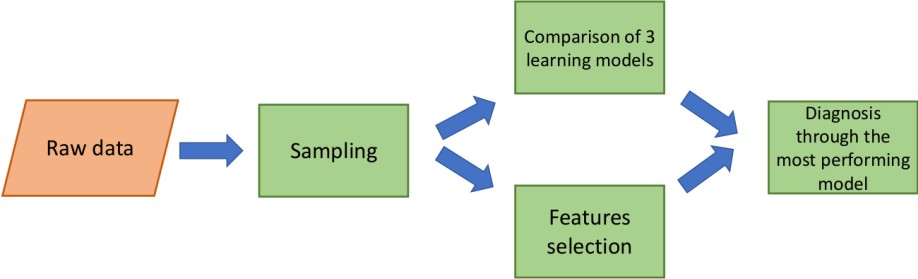

**Figure 1.** Flowchart of the proposed methodology. After sampling we selected the most performing learning algorithm and implemented a feature reduction procedure. Finally we used the selected learning model to classify the patients in the three CVD risk classes.

A detailed description of these steps is provided in the following sections.

### 2.2.1. Data Sampling

First of all, since the data used in this work are characterized by high heterogeneity, the features have wide and poorly comparable range of values; accordingly, a data standardization was performed as a preliminary in which we rescaled data to have a mean of 0 and a standard deviation of 1. For the modelling phase, the data presented a typical issues of a learning analysis—data imbalance. A dataset is balanced if the classes are approximately equally represented. Learning algorithms require a balanced dataset in order to yield accurate predictions, which, otherwise, would be severely biased towards the majority class. The data imbalance can be solved using two opposite strategies: on the one hand, it is possible to over-sample the classes less populated; on the other hand it is possible to under-sample the majority class. Every strategy has its own advantages and drawbacks, for the present case we chose an oversampling technique because of the limited available sample size.

In particular, in this work we used the Synthetic Minority Over-sampling Technique (SMOTE) sampling method. This technique creates new instances of the minority classes from combinations of preexisting instances. A particular advantage of such an approach is to create novel synthetic examples rather than using duplicates, which is the case of other more standard approaches such as bootstrap. A peculiar aspect of SMOTE is that it generates synthetic examples by operating in feature space rather than data space [37]. To over-sample the minority class, each minority class observation is taken into account and its $n$ minority-class neighbors are also considered; then, new synthetic examples are introduced along the directions connecting the considered example and its neighbors, thus resulting in $n$ novel synthetic observations.

### 2.2.2. Feature Selection

After having the data standardized and the three classes balanced, we performed a feature selection procedure based on the Boruta algorithm [38]. Boruta is a wrapper method that exploits the Random Forest classifier to perform a robust and efficient supervised feature selection. Given the relatively small number of features, we might not even apply a feature selection strategy, but through Boruta we reduce the noise or redundant data, thus we can improve our model by using only those features that are uncorrelated and non-redundant. In fact, a wrapper method like Boruta plays an important role not only reducing the computational time of the algorithm but decreasing the complexity of the model and improving the model performances. Briefly, Boruta (Boruta is a god of the forest in the Slavic mythology) exploits the same idea behind the Random Forest classifier, that is, by inserting elements of randomness into the system and calculating results from the set of randomized samples, it is possible to reduce the negative effect of random fluctuations and correlations [38]. In Boruta, Random Forest trees are independently grown on different bagging samples of the training set. Then, feature importance is computed through the loss of classification accuracy caused by random permutation of the variables.

### 2.2.3. Classification Model

Parallel to the feature selection procedure, we compared the classification performances of three different algorithms: General Linear Model, Random Forest and Neural Networks. These models have been fed with the 52 considered features (27 heavy metals concentration + 25 polymorphisms). We started with a linear hypothesis and then we applied machine learning models to improve the classification performances. To increase the robustness of our findings, we used two very different machine learning algorithms as neural networks based on the artificial neuron as fundamental unit and Random Forest composed of an ensemble of decision trees. These algorithms are very versatile and not require any a priori assumption.

The Generalized Linear Model (GLM) expands and, in a sense, completes the perspective and the applications field of the linear regression model [39]. It was formulated by John Nelder and Robert Wedderburn as a way to uniform different statistical models within a single model including the linear model, logistic regression and Poisson regression [40]. While in the classical linear model it is assumed that the dependent variable has a normal distribution, in the GLM the dependent variable can be distributed as any variable belonging to exponential family: binomial, poissonian, gamma, etc. . . The linear model presumes that the expected value of the dependent variable $y$ is provided by a linear combination of the independent variables $x$. This introduces a limitation of linearity that restricts the practical application field. Instead, GLM introduces a linearizing link function which transforms the expectation value of the dependent variable [41]. In this way, non-normal and discrete distributions of $y$ can also be described through this model [42]. Specifically, GLM is composed of three components [43]:

1. A random component that specifies the conditional distribution of the dependent variable $y = y_i, \ldots, y_n$ composed of $n$ independent observations in relation to the values of the independent variables of the model.
2. A linear function of regressors.

$$\nu_i = \alpha_1 x_{i1} + \alpha_1 x_{i2} + \ldots + \alpha_1 x_{ik} + \gamma \tag{1}$$

3. A linearizing link function $g()$ that converts the expectation of $y_i$, $\chi_i$ in $\nu_i$.

$$g(\chi_i) = \nu_i = \alpha_1 x_{i1} + \alpha_1 x_{i2} + \ldots + \alpha_1 x_{ik} + \gamma \tag{2}$$

Random Forest (RF) is a machine learning algorithm which exploits a set of decision trees built through resampling with repetitions (bootstrapping) of the training dataset [44]. In the training phase, at each node of a RF tree, a subset of features is randomly selected so that the RF trees are poorly correlated with each other. Then, the classification task is performed through a majority vote procedure; each tree yields its own prediction and the final decision is obtained by majority voting.

In general, RF has some features that make it a suitable choice for many classification problems: (i) RF is easy to tune with only two parameters to tune, the number of trees $t$ used to grow the forest and the number of features $m$ sampled to grow each leaf within a tree; (ii) it is robust to overfitting; (iii) using an out-of-bag procedure, RF can compute an unbiased estimation of the generalization error. Node impurity is measured by the Gini index, in supervised classification tasks, this metric allows the forest to learn the optimal cuts and how to optimally separate the available classes [45]. In our analysis, in order to obtain results independent from the model tuning, we adopted a standard configuration with each forest grown with $t = 300$ trees and $m = f/3$ ($f$ being the total number of features).

Artificial neural networks (ANNs) are constituted by computational networks reproducing the human nervous system that can learn from known examples and generalize to unknown cases. In our work, we implemented multilayer perceptron networks (MLPs) [46], the most commonly used ANNs, that used the back-propagation algorithm in the supervised learning process. The MLP's architecture provides three neural levels: input, hidden and output layers. In the learning process a features array feeds the input layer. Then the input flows to the next hidden layers through dendrites, i.e., neural connections; during the crossing of the network, the signal can be inhibited or amplified through the connection weights and the neurons add the input signals and transform them into output signals by means of an activation function. Our MLP model was composed of one hidden layers with 3 neurons and used the sigmoid as an activation function.

### 2.2.4. Cross-Validation and Performance Metrics

In order to estimate the goodness and the robustness of our classification model a cross validation framework was applied. In general, for a $k$-fold Cross Validation (CV) procedure, the original dataset is randomly partitioned into $k$ equal sized subsamples. Of the $k$ subsamples, a single subsample is retained for validation and the remaining $k - 1$ subsamples are used to train the model. To select the most performing algorithms between GLM, RF and ANNs we applied a common five-fold. Then we used the feature set individuated by Boruta to feed this learning model and to distinguish the three-levels CVD risk. To make our model more robust we studied the trend of the classification accuracy while varying the k parameter: $k = 10, 5, 3$ and Leave One Out (LOO) Cross Validation. However, for the present case such procedure could greatly affect the model accuracy. Accordingly, we also assessed a LOO cross validation in order to maximize the number of observations used for training. It is well known that the results obtained with such procedure should be considered cum grano salis as they could lead to an accuracy overestimation. A figurative example of our procedure is shown in Figure 2.

The model performance were firstly evaluated in terms of Area Under Roc Curve (AUC) and accuracy, namely the rate of correct classifications. In a classification problem with $M$ classes, the accuracy for the class $i$, with $i = 1, \ldots, M$ is defined as:

$$Acc_i = \frac{TP_i + TN_i}{TP_i + TN_i + FP_i + FN_i} \tag{3}$$

where $TP_i, TN_i, FP_i, FN_i$ represent true positives, true negatives, false positives and false negatives, respectively, for $i - th$ class. The overall accuracy is obtained from the average on all classes. In addition, we also evaluated the model sensitivity and precision and the F1 score for each class $i$:

$$Sens_i = \frac{TP_i}{TP_i + FN_i}; \tag{4}$$

$$Prec_i = \frac{TP_i}{TP_i + FP_i}; \tag{5}$$

$$F1_i = 2 \cdot \frac{Prec_i \cdot Sens_i}{Prec_i + Sens_i}. \tag{6}$$

All the data processing and the carried out statistical analyses were performed in R version 4.1.0 (https://www.r-project.org/, accessed on 31 August 2021).

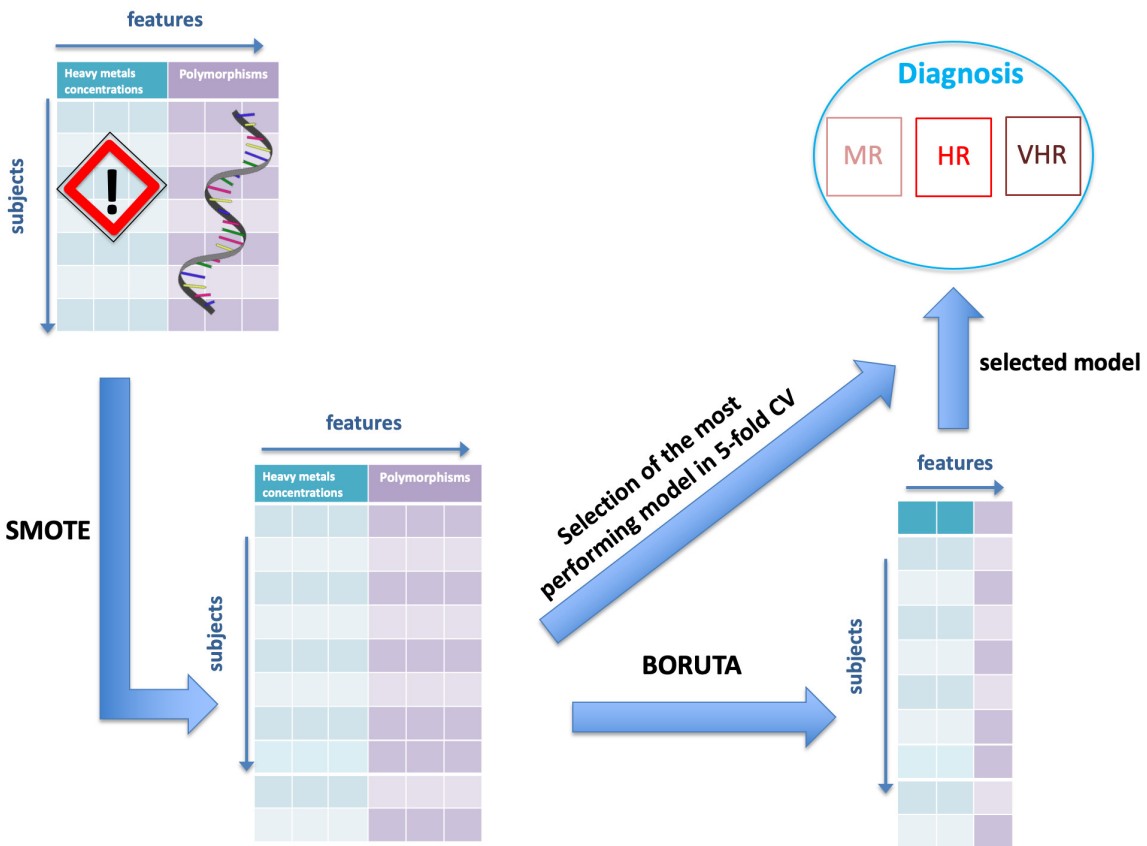

**Figure 2.** Pictorial representation of the implemented classification procedure. After balancing the dataset through SMOTE method, we selected the most efficient feature subset by means of Boruta algorithm and the most performing learning model in 5-fold CV. We used the chosen feature array to feed the selected classifier and distinguish the three-levels CVD risk.

## 3. Results

### 3.1. Comparing Different Machine Learning Strategies

First of all, we evaluated to which extent heavy metals concentrations and SNPs provide a valuable tool to predict the severity of CVD conditions. To this aim, we performed a repeated (100 times) 5-fold cross-validation and compared the performance of GLM, RF and ANN; see Figure 3.

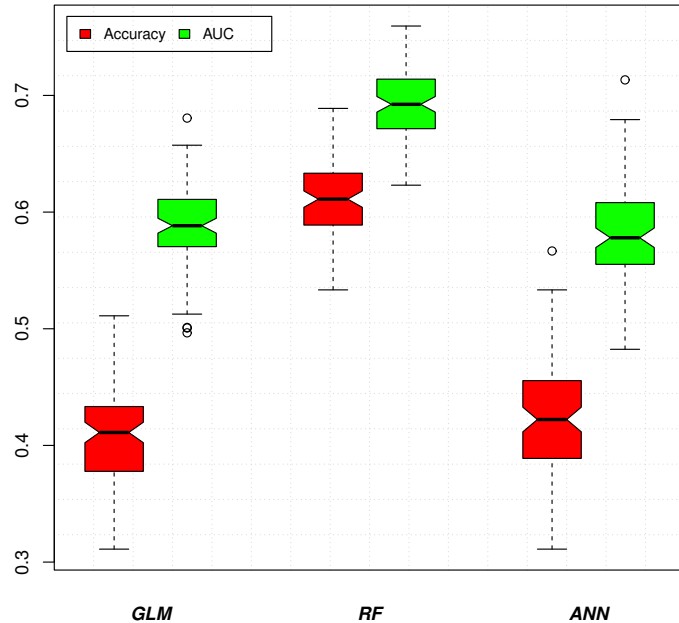

**Figure 3.** Accuracy and AUC for the three implemented models: GLM, RF and ANN. Each distribution was computed through a 5-fold cross validation procedure repeated 100 times. Empty bullets represent the outliers of distributions.

RF resulted the best performing method in term of accuracy ($0.61 \pm 0.03$) and AUC ($0.69 \pm 0.03$). GLM and ANN obtained accuracy values of $0.41 \pm 0.04$ and $0.42 \pm 0.05$, and AUC equal to $0.59 \pm 0.04$ and $0.58 \pm 0.04$, respectively. For further detail, other metrics for each class of each model are reported in the following Table 3.

**Table 3.** Sensitivity, precision and F1 score of the three models GLM, RF and ANN for each class. The classification performances are reported with the respective standard deviations.

| | Sensitivity | | |
|---|---|---|---|
| **Class** | **GLM** | **RF** | **ANN** |
| MR | $(43.35 \pm 6.62)\%$ | $(63.21 \pm 5.60)\%$ | $(41.77 \pm 8.74)\%$ |
| HR | $(30.00 \pm 10.27)\%$ | $(47.94 \pm 7.88)\%$ | $(36.00 \pm 11.80)\%$ |
| VHR | $(44.26 \pm 6.46)\%$ | $(65.21 \pm 4.68)\%$ | $(45.63 \pm 7.88)\%$ |
| | **Precision** | | |
| **Class** | **GLM** | **RF** | **ANN** |
| MR | $(49.34 \pm 6.64)\%$ | $(59.12 \pm 4.19)\%$ | $(47.35 \pm 6.19)\%$ |
| HR | $(19.88 \pm 5.68)\%$ | $(45.16 \pm 6.80)\%$ | $(23.54 \pm 7.01)\%$ |
| VHR | $(50.18 \pm 5.80)\%$ | $(72.30 \pm 4.13)\%$ | $(53.59 \pm 6.51)\%$ |
| | **F1** | | |
| **Class** | **GLM** | **RF** | **ANN** |
| MR | $(45.36 \pm 5.86)\%$ | $(60.99 \pm 4.18)\%$ | $(44.11 \pm 7.09)\%$ |
| HR | $(23.77 \pm 7.13)\%$ | $(46.28 \pm 6.66)\%$ | $(28.27 \pm 8.43)\%$ |
| VHR | $(46.87 \pm 5.56)\%$ | $(68.47 \pm 3.62)\%$ | $(49.02 \pm 6.43)\%$ |

In terms of classification performance, according to a Kruskal–Wallis test [47], no significant difference among tested methodologies were detected.

Besides, we evaluated the consistency of predictions of all three classifiers through pairwise contingency tables, see Figure 4.

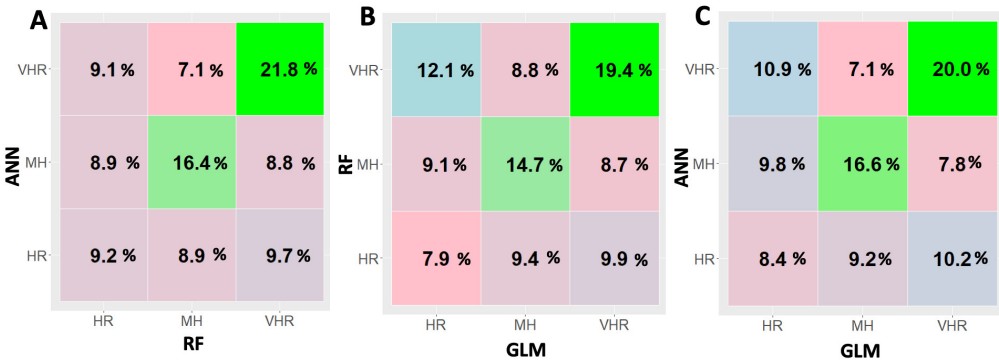

**Figure 4.** Contingency tables with pairwise comparisons of the implemented algorithms predictions: RF vs. ANN (panel **A**), RF vs. GLM (panel **B**), ANN vs. GLM (panel **C**). The value are averaged over 100 rounds of 5-fold cross validation.

It is worth noting that, in all three cases, the agreement between the classification models exceeds the 42%. Accordingly, for further analyses only RF models were taken into account.

### 3.2. Clinical Features Affecting CVDs

A additional goal of this study was to evaluate which were the clinical features mostly affecting the CVD severity. For this purpose, we performed a feature importance analysis. Starting from the 52 initial features, through Boruta algorithm we selected the features maximizing the classification performance, as shown in Figure 5.

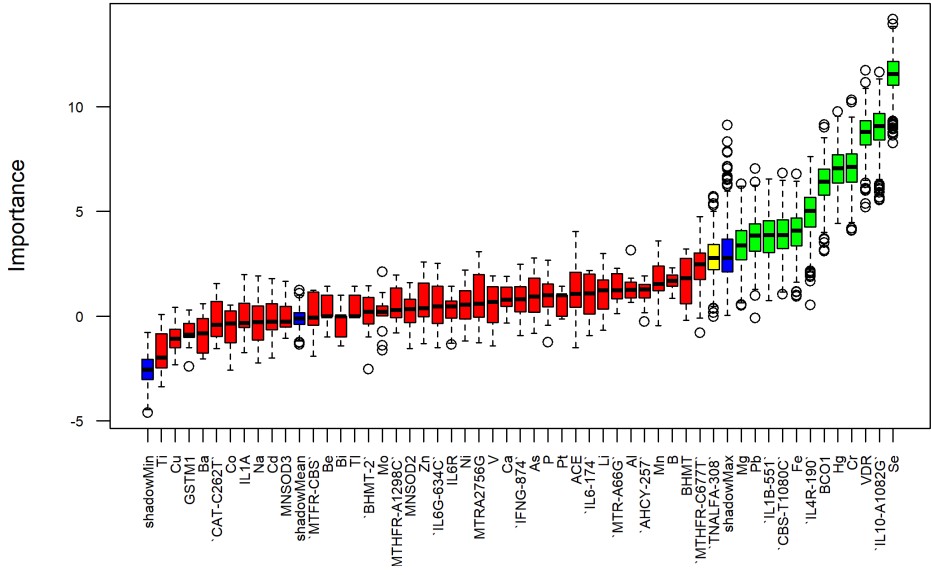

**Figure 5.** Boxplots of attribute importance defined by Boruta's algorithm. The wrapper algorithm selects 13 important features (in green). The importance is computed in term of Z-score. Shadow Boruta predictors are shown in blue. Red colors indicate unimportant features, and yellow colors tentative attributes, i.e., features that are almost as important as their best shadow attributes and Boruta cannot make a decision about them with the desired confidence. Empty bullets represent the outliers of distributions.

The figure shows how 13 features were recognized as important for the classification. Among the chosen features, 6 heavy metals and 7 SNP were selected; the list is provided in Table 4.

**Table 4.** List of the selected features. The feature are achieved by means of Boruta algorithm.

| Selected Metals | Selected Polymorphisms |
|:---:|:---:|
| Se | IL10 A1082G |
| Cr | VDR |
| Hg | BCO1 |
| Fe | IL4R 190A/G |
| Pb | IL-1βC-551T |
| Mg | CBS T1080C |
| | TNF α/G-308A |

These results suggested to consider a restricted number of features to possibly improve the classification performance.

### 3.3. Assessing the CVD Severity

Finally, we used the 13 features selected by Boruta to feed RF classifier and investigate if any classification performance was achieved; see Figure 6.

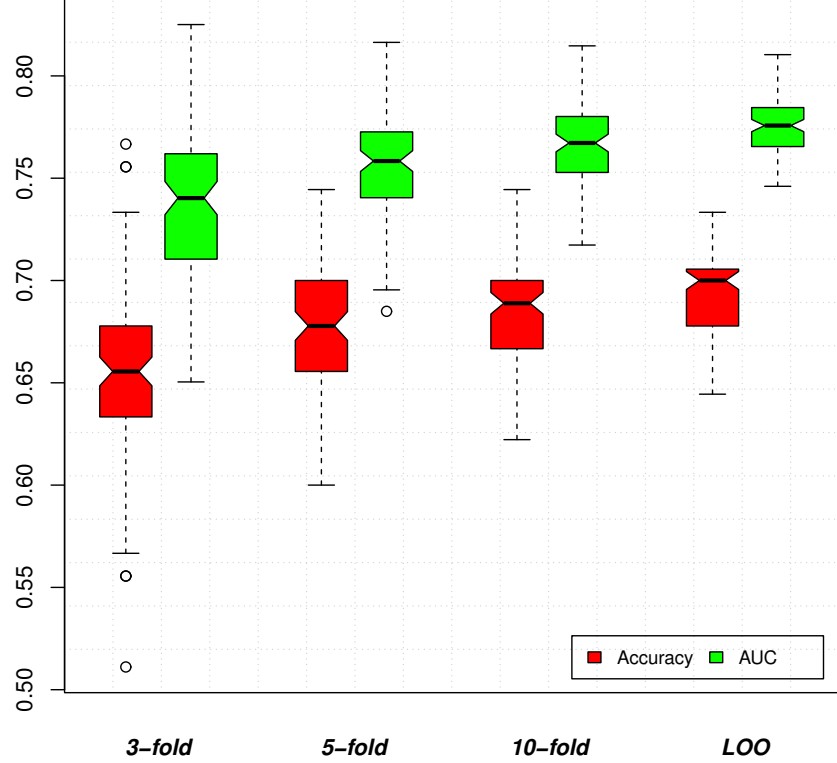

**Figure 6.** Accuracy and AUC values for different cross validation frameworks. All CV frameworks 100 times have been repeated. Empty bullets represent the outliers of distributions.

These results show how the selected features improve significantly the classification performances. For 5-fold CV procedure the accuracy value goes from $0.61 \pm 0.03$ to

$0.68 \pm 0.03$. Furthermore, the AUC value increases from $0.69 \pm 0.03$ to $0.75 \pm 0.03$. To evaluate the effect of sample size over the reported performance, we also compared different cross-validation schemes. The best performance was obtained with a LOO framework, in this case maximum values of both accuracy ($0.69 \pm 0.02$) and AUC ($0.78 \pm 0.01$) were reported; nevertheless, no significant differences arise with other cross-validation schemes.

In Table 5, the mean sensitivity, precision and F1 values computing through a LOO framework for the three risk classes are reported.

**Table 5.** Summary table of sensitivity, precision and F1 measures for the three considered classes. The classification metrics were obtained with RF and were averaged over 100 rounds of cross-validation. All values are reported with the relative standard deviations.

| Class | Sensitivity | Precision | F1 |
|---|---|---|---|
| Moderate risk | $(77.44 \pm 3.04)\%$ | $(65.71 \pm 2.15)\%$ | $(71.08 \pm 2.24)\%$ |
| High risk | $(46.78 \pm 3.88)\%$ | $(48.96 \pm 3.41)\%$ | $(47.79 \pm 3.27)\%$ |
| Very high risk | $(72.94 \pm 1.68)\%$ | $(84.84 \pm 2.47)\%$ | $(78.41 \pm 1.44)\%$ |

*3.4. Heavy Metals vs. SNPs Informative Power*

As the feature importance analysis showed that both heavy metals and SNPs contributed to best discriminating the CVD severity, we finally investigated the informative power of these features, separately. To this aim, we trained two RF models, the first one with the 27 heavy metals concentrations and the second one with the 25 SNPs. We compared these models with the model obtained using the 13 features selected by Boruta; see Figure 7.

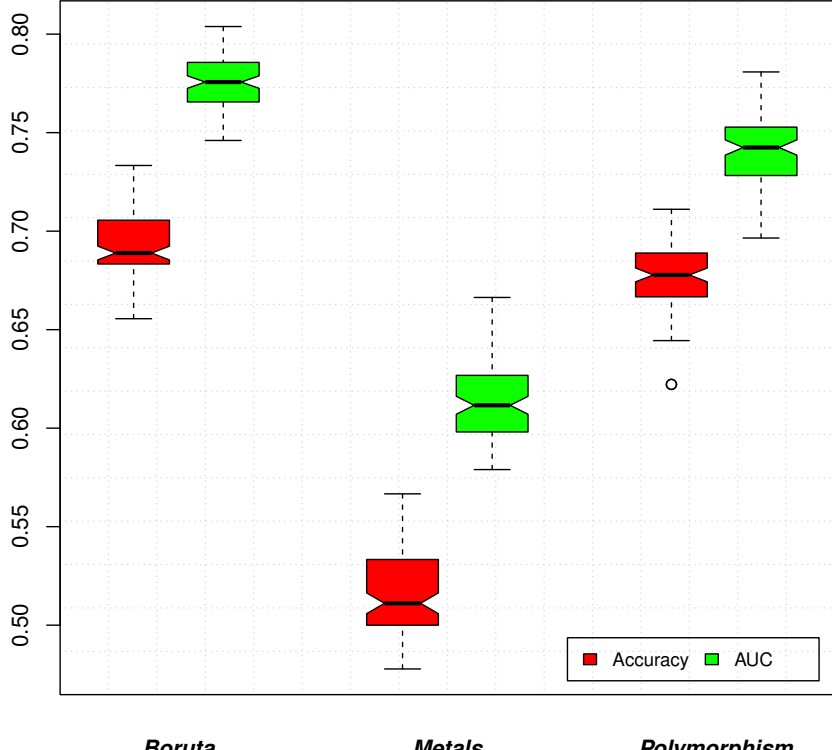

**Figure 7.** Accuracy and AUC values for 3 different RF models. The first classifier has been fed with the features selected by Boruta; the second with 27 heavy metals concentrations; the third with 25 polymorphisms. Each distribution was computed through a LOO cross validation procedure repeated 100 times. Empty bullets represent the outliers of distributions.

Boruta features yielded the most accurate model. Although SNPs provide an excellent base of knowledge, compared with the discriminating power of heavy metals concentrations, these results show that the combined use of both concentrations and genetic polymorphisms significantly (*p*-value < 0.05 whatever test you used) improves the classification results.

## 4. Discussion

The presence of heavy metals in the environment where humans live, represents a factor of great importance in the onset of degenerative pathologies (cardiovascular and neurodegenerative diseases, neoplasia, etc...) as clearly demonstrated by a lot of scientific works. For example, Buonovato et al. by means of scalp hair analysis implemented in the industrial city of Taranto reported an increase of lung cancer risk among residents due to the high concentration of heavy metals [48]. In the present work, we focused on the relation between heavy metals and cardiovascular diseases. We applied a learning methodology based on Artificial Intelligence algorithm to analyze a private database containing 27 heavy metal concentrations and 25 polymorphisms linked to the cardiovascular diseases of 90 patients enrolled by DD Clinic foundation of Caserta in Italy. Through a machine learning procedure based on three state of the art algorithms, we classified patients with cardiovascular diseases in three classes of risk. Furthermore, we adopted a sampling strategy based on SMOTE method to allow the best operational conditions for each classifier. The best-performing method was RF in terms of all used classification metrics as shown in are shown in Figure 3 and in Table 3. We observed that, for all pairwise comparisons, the agreement between the classification models is not very high (42–46%). We investigated, by means of the Boruta wrapper method, which features were best at characterizing the differences between the three levels of cardiovascular risk. We observed that a small amount of features (25% of the initial dataset, as reported in Table 4) was sufficient for an accurate classification. Furthermore, we adopted different cross validation frameworks to maximize the performance of each RF classifier. We obtained good classification performances (0.78 of AUC), considering a three classes problem, when we used RF fed by the Boruta features in LOO cross validation, as shown in Figure 6 and in Table 5. The discriminating power of these selected features is higher than that of the entire set of features; but also of the heavy metal concentrations and of the polymorphisms taken separately, as displayed in Figure 7. This means that Boruta has efficiently eliminated noise and redundant feature. The use of Boruta is also important to assess the importance of different predictors and thus provide clinical interpretable findings. Furthermore, the results summarized in Figure 7 show how the implemented wrapper method strategy improves the classification performance. Although the size of input data could appear small for a machine learning approach, our findings show RF, with SMOTE, Boruta and cross validation, is a robust framework for CVD classification. However the application of an oversampling technique does not solve completely the data imbalance issue. In fact, the minority class HR presents ever the worst performances. Our results show that the starting hypothesis that heavy metal concentrations and SNPs can accurately model the severity of CVD was correct. However the selected features displayed in Table 4 testify that the association between environmental factors and CVD happens by means of complex biological and genetic pathways. The presence of these pathways was highlighted thanks to a machine learning approach and therefore without a priori assumptions about the relationship between the input features and the severity of CVD. These pathways should be studied further to understand how heavy metals affect cardiovascular disease.

To confirm the goodness of our findings in previous works, some of the selected features had already been highlighted as linked to cardiovascular diseases. For example, Chowdhury et al., by analyzing 37 unique studies comprising 348259 participants, concluded that exposure to lead (as well as to arsenic, cadmium, and copper) is associated with an increased risk of cardiovascular disease and coronary heart disease [49]. Batuman et al. found that lead can have an important function in the renal disease of a cohort of

patients usually designated as suffering from "essential" hypertension [50]. Furthermore, the harmful effects due to a high concentration of lead, even in infancy, are well established and documented [51–53].

The correlation between iron accumulation in the body and the onset of heart and cardiovascular diseases has also been widely studied. Sullivan states that the greater occurrence of heart illnesses in men and postmenopausal women respect to the incidence in premenopausal women is due to higher amount of stocked iron in these two groups [54]. He also discovered that body iron stores are directly linked to coronary heart disease risk, i.e., the greater the accumulation of iron the greater the risk [55]. These results have been confirmed by Salonen et al, which found that men with levels of serum ferritin (a measure of body iron stores) higher than 200 μg/L is subjected to double the risk of having a heart attack [56]. In addition, it is known that an excessive accumulation of iron could induce atherosclerosis by acting as a catalyst for oxidation of low-density lipoproteins (LDL) cholesterol [57]. Many works claimed that the mercury has no known physiological role in human metabolism and its high concentration is strongly related with hypertension, coronary heart disease, myocardial infarction, cardiac arrhythmias, carotid artery obstruction and cerebrovascular accident [58,59]. Moreover it represents a risk factor in the atherosclerosis progression [60]. Selenium, and magnesium also appear to play an important role in the development of cardiovascular disease, but in this case the effect appears to be reversed. In fact, some authors assumed a possible protective role of these elements against CVD.

There is a lot of evidence about the importance of selenium and its selenoproteins in the cardiovascular system, due to its well-known antioxidant characteristics [61]. A study conducted by Flores-Mateo et al. on a dataset of 25 patients discovered that *Se* concentrations were inversely associated with coronary heart disease risk [62].

Some polymorphisms of genes highlighted through our feature selection procedure have already been implicated in the progression of CVD in other studies. SNPs of a gene involved in the vitamin D cascade as VDR could be risk factors for acute coronary and cardiovascular syndrome [63,64]. It is known that BCO1 is a gene implicated in carotenoid metabolism and Cai et al. demonstrated that the associated polymorphisms are realated to the risk of coronary atherosclerosis [65]. TNF alpha/G-308A polymorphism might interact with some factors linked to coronary heart disease to induce an insulin resistance [66]. A research conduced by Ma et al. on over 1000 subjects found that mutation at the $IL - 1\beta$ promoter region ($C - 511T$) predisposed in diabetic patients the development of coronary artery diseases evidence [67].

In general, our findings would suggest that our machine learning tool can also be effectively used to assess the CVD risk although deeper validation is needed on a larger sample of patients.

## 5. Conclusions

It is well known that high concentrations of heavy metals may be harmful to human health and are related to the onset of degenerative pathologies such as cardiovascular pathologies, neoplasia and neurodegenerative diseases, just to mention a few. Human scalp hair analysis (HSHA) is a minimally invasive test that detects the toxic minerals in the human body with high accuracy. In this work, we designed a machine learning framework to assess the cardiovascular diseases risk from HSHA tests and genetic analysis and investigate which clinical features (heavy metal concentrations and SNPs) are mostly associated with the CVD severity. We implemented and compared different cross-validation schemes and machine learning strategies to grant the robustness of our findings. Using private cohort of 90 patients affected by CVD with different levels of severity, we observed that heavy metals and SNPs can achieve an AUC of $0.78 \pm 0.01$ on a three classes classification problem. Although the clinical role of the best performing features, among the 57 adopted here, detected in our analysis have been partially outlined in other CVD studies, this is the first study to emphasize their combined importance. In this perspective, our findings may

be useful to shed further light over the underlying molecular mechanisms of CVD and eventually identify possible targets for the development of novel therapies and drugs [68].

**Author Contributions:** Conceptualization, A.M. and A.I.G.; methodology, A.L., A.M., N.A. and A.I.G.; software, A.L.; formal analysis, A.L.; writing–original draft preparation, A.M. and N.A.; writing–review and editing, all the authors; visualization, A.L. and A.M. All authors have read and agreed to the published version of the manuscript.

**Funding:** This research received no external funding.

**Institutional Review Board Statement:** Ethical approval was not required because the activities described were conducted as part of routine diagnostic tests. All procedures were carried out in accordance with the Declaration of Helsinki, as revised in 2013, for research involving human subjects. The data were deidentified; therefore, the need for informed consent was waived.

**Data Availability Statement:** Data analysed during the current study is available at the the following link: https://github.com/amonaco3280/Environmental-heavy-metals, accessed on 31 August 2021.

**Conflicts of Interest:** The authors declare no conflict of interest.

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
