# Peer review of "Random Forests Highlight the Combined Effect of Environmental Heavy Metals Exposure and Genetic Damages for Cardiovascular Diseases"

_applsci, doi:10.3390/app11188405_

Round 1

Reviewer 1 Report

This manuscript reports the outcome of a research endeavour to relate cardiovascular disease severity and genetic analyses results with heavy metal concentrations using a random forest model. Its ultimate objective is finding insights with clinical relevance.

The title is too general. It is not sufficiently specific or informative, and it can mislead readers by alluding to aspects not covered in the article. Which Machine Learning approaches and techniques? Which heavy metals? Under what conditions? 

The abstract is too long and has an unnecessarily long introductory part. It should go straight to the point.

The research is not framed into the current body of knowledge. To such an end, Authors would need to engage, at least, in a brief literature review covering the aspects they are dealing with and its current research status. However, such a review has not been made, either in a Literature Review chapter or within the Introduction.

Interestingly, there is a brief review in the Discussion section.

Lacking the research status, it was not possible to accurately depict the research problem and the research gap. Therefore, the purpose of this research and its possible impact upon the current knowledge in its field are not shown.

The theme is current and interesting.

Research methods are adequate. I was not able to find any novelty within the employed computer science, but it is undoubtedly well-known and tested through the years.

Data has not been made available. Therefore the study is not replicable. Under these conditions, there is not much to be said pertaining to its scientific soundness. Furthermore, its conclusions acceptance will depend on the readers' trust in the study. This is usually considered unsatisfactory for machine learning applications.

Discussion should focus on the attained results, with an in-depth analysis. That is not the case in this manuscript.

Conclusions are adequate.

The document is generally well written and organized, but one can find some typos and informal language. A full revision shall be made, but some specific issues can be found in "presented a major issues" or "an enormous amount of scientific literature".

Author Response

Please find in attachment the answers to the points raised by the reviewer.

Reviewer 2 Report

Thanks for recommending me as a reviewer. In this paper, the authors proposed a machine learning framework to evaluate cardiovascular diseases (CVD) severity from Human scalp hair analysis (HSHA) tests and genetic analysis and identify a small group of these clinical features mostly associated with the CVD risk. If the authors complete the revision, the quality of the study will be further improved.

1.The introduction section is well written. If the authors describe in more detail the necessity and theoretical background of machine learning to identify the risk factors of CVD in the introduction section, it can help the reader's understanding.

2. line 98-103: Authors should be more specific about the source (“Ambulatorio Fondazione DD Clinic”) and subjects in the Methods section.

3. The authors have well described methods related to feature selection (eg SMOTE) in the Methods section. It may be helpful for readers to understand if the authors describe in more detail what variables (features) are included in the data source. Authors can present a list of variables included in the analysis in a table form.

Author Response

(The authors gave the same response as above.)

Round 2

Reviewer 1 Report

Most of my comments have been successfully addressed by the Authors, including the critical one concerning data availability.